# Real-Time Motion-Controllable Autoregressive Video Diffusion

**Kesen Zhao**[1]* **Jiaxin Shi**[2]† **Beier Zhu**[3]‡ **Junbao Zhou**[1] **Xiaolong Shen**[4] **Yuan Zhou**[1]
**Qianru Sun**[5] **Hanwang Zhang**[1]
[1]Nanyang Technological University, [2]Xmax.AI Ltd, [3]University of Science and Technology of China
[4]Zhejiang University, [5]Singapore Management University
`kesen002@e.ntu.edu.sg`

## Abstract

Real-time motion-controllable video generation remains challenging due to the inherent latency of bidirectional diffusion models and the lack of effective autoregressive (AR) approaches. Existing AR video diffusion models are limited to simple control signals or text-to-video generation, and often suffer from quality degradation and motion artifacts in few-step generation. To address these challenges, we propose `AR-Drag`, the first RL-enhanced few-step AR video diffusion model for real-time image-to-video generation with diverse motion control. We first fine-tune a base I2V model to support basic motion control, then further improve it via reinforcement learning with a trajectory-based reward model. Our design preserves the Markov property through a Self-Rollout mechanism and accelerates training by selectively introducing stochasticity in denoising steps. Extensive experiments demonstrate that `AR-Drag` achieves high visual fidelity and precise motion alignment, significantly reducing latency compared with state-of-the-art motion-controllable VDMs, while using only 1.3B parameters. Additional visualizations can be found on our project page: `https://kesenzhao.github.io/AR-Drag.github.io/`.

## 1 Introduction

Video diffusion models (VDMs) have made remarkable progress with bidirectional diffusion transformers (DiTs), which denoise all frames simultaneously (Kong et al., 2024; Villegas et al., 2022; Wan et al., 2025; Yang et al., 2024; Wang et al., 2025a). As shown in Fig. 1 (a), they allow future information to influence the past and require generating the entire video frames jointly. All existing motion-controllable VDMs are dominated by this bidirectional design. As a result, generation is delayed until all control inputs are specified, causing high latency and disallowing real-time adjustment of controls, *e.g.*, sequential motion cues that evolve as the video unfolds. In contrast, autoregressive (AR) VDMs (Yin et al., 2025; Gao et al., 2024; Gu et al., 2025; Lin et al., 2025) generate videos sequentially, making them naturally aligned with real-time controllable video generation.

Despite being well-suited to real-time control, existing AR VDMs primarily target text-to-video (T2V) generation and remain limited in the more challenging image-to-video (I2V) scenarios (Yin et al., 2025; Huang et al., 2025), or only explore simple control signals such as pose or camera motion (Lin et al., 2025; Shen et al., 2024). Controllable AR VDMs face two major challenges: **(1)** quality degradation and motion artifacts caused by error accumulation, especially for few-step models. **(2)** richer control modalities such as trajectories or bounding boxes (Zhang et al., 2025), that broaden the action space and require stronger generalization. To the best of our knowledge, our `AR-Drag` (Fig. 1(b)) is the first AR VDM enabling real-time motion control with visual quality competitive with bidirectional ones. As shown in Fig. 1(c), `AR-Drag` achieves substantially lower latency while maintaining superior FID compared with state-of-the-art motion-controllable VDMs.

---

*Work was done during an internship at Xmax.AI.

†Project leader.

‡Corresponding author.

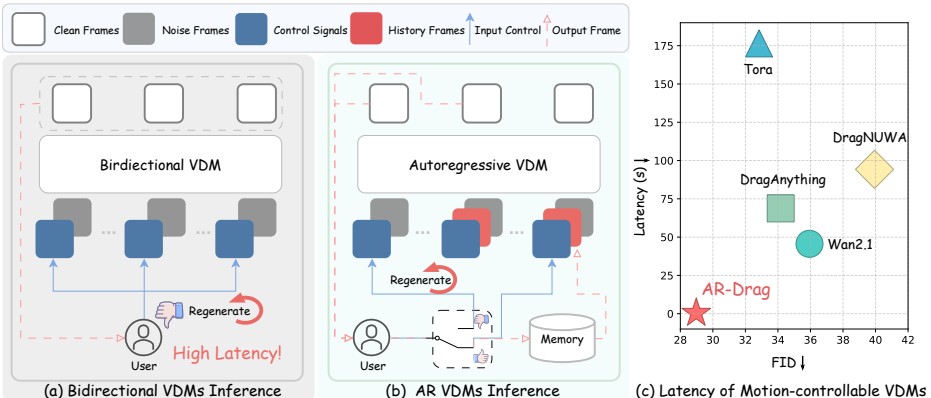

Figure 1: Comparison for motion-controllable video generation. (a) Bidirectional VDMs denoise all frames jointly; motion control can be adjusted only after all frames are generated, causing high latency. (b) In contrast, AR VDMs generate frames sequentially; motion control can be updated frame by frame and, if unsatisfactory, regenerated on the fly, enabling real-time adjustment. (c) Our method achieves significantly lower latency while maintaining superior FID performance.

In response to the two challenges, reinforcement learning (RL) is a natural fit. Unlike supervised learning, which enforces pixel-level reconstruction and limits the model to the training distribution, RL explores the action space and optimizes policies via trial-and-error, enabling strategies that generalize beyond seen data. Recent work built on GRPO (Guo et al., 2025), such as DanceGRPO and FlowGRPO (Xue et al., 2025; Liu et al., 2025), demonstrates the effectiveness of RL for bidirectional flow-matching models in text-to-image (T2I) generation. However, extending GRPO to video generation raises several challenges: (1) ensuring the Markov property, since typical AR VDMs condition on ground-truth frames during training rather than self-generated ones, breaking the MDP formulation; (2) handling the long decision process of video generation, where exploration across the entire decision chain becomes prohibitively expensive; (3) the lack of well-defined reward models tailored to controllable video generation.

To address these issues, we propose `AR-Drag`, an RL-enhanced few-step AR VDM for real-time motion-controllable I2V generation. Specifically, we first fine-tune the Wan2.1-1.3B (Wan et al., 2025) I2V model on our curated control-aware data to enable basic motion control, and then further improve it through reinforcement learning. To preserve the Markov property, we introduce **Self-Rollout**, training on model-generated histories to align with AR inference. To keep long-horizon exploration tractable, we adopt **selective stochasticity**: a single randomly chosen denoising step uses an SDE update, while all remaining steps follow the deterministic ODE solver. In addition, we design a trajectory-based reward model to enforce fine-grained control over complex motion signals.

Our contributions are threefold: (1) We propose `AR-Drag`, the first few-step AR VDM capable of real-time controllable I2V generation. (2) We introduce RL-based training for AR VDM and design a trajectory-based reward model tailored to fine-grained motion alignment. (3) We conduct extensive experiments showing that `AR-Drag` significantly improves both visual quality and controllability, despite using only 1.3B parameters.

## 2 RELATED WORKS

**Controllable video generation.** Textual modality have been extensively studied (Zhu et al., 2025b; 2024b;a), whereas motion control in VDMs remains relatively underexplored. Early methods (Jeong et al., 2024; Wang et al., 2023; Zhao et al., 2024) achieve motion control by injecting motion signals into VDMs, yet their capability is restricted to reproducing pre-defined dynamics. Recent works (Geng et al., 2025; Ma et al., 2024; Mou et al., 2024; Shi et al., 2024; Wang et al., 2024; Yin et al., 2023; Zhang et al., 2025; Wu et al., 2024; Wang et al., 2025b; Zhou et al., 2025) leverage explicit control inputs such as motion trajectories, offering greater flexibility. For example, DragAnything (Wu et al., 2024) leverages object masks for entity-level control, and Tora (Zhang et al., 2025) introduces trajectory conditioning into a DiT framework. However, all these methods are non-autoregressive and therefore unsuitable for real-time interactive control.

**Real-time video generation.** Video diffusion models typically adopt bidirectional attention mechanism (Blattmann et al., 2023a;b; Brooks et al., 2024; Ho et al., 2022; Kong et al., 2024; Villegas et al., 2022; Wan et al., 2025; Yang et al., 2024). While effective for quality, this design requires jointly denoising all frames of video, limiting their applicability to real-time interactive. Autoregressive models (Hu et al., 2024; Jin et al., 2024; Yin et al., 2025; Gao et al., 2024; Gu et al., 2025; Li et al., 2025c), in contrast, generate tokens sequentially, making them inherently better suited for real-time controllable video generation. Some attempts (Yin et al., 2025; Lin et al., 2025; Yang et al., 2025) distill multi-step VDMs into few-step autoregressive VDMs using distribution matching distillation (Yin et al., 2024b;a) or consistency distillation (Song et al., 2023; Song & Dhariwal, 2023). However, AR VDMs still exhibit a train–test mismatch, making them prone to error accumulation across frames—particularly in few-step models. To mitigate this, some works (Chen et al., 2024; Teng et al., 2025; Sun et al., 2025) propose progressive noise schedules that gradually increase noise from early to later frames, partially alleviating error accumulation. However, they neither close the train–test gap nor support real-time interaction, since future frames must be pre-generated before the current frame is rendered, introducing latency and limiting control effectiveness. Self-Forcing (Huang et al., 2025) narrows the train–test gap and improves stability by unrolling autoregressive generation during training, conditioning each frame on previously generated outputs rather than ground truth. However, it does not strictly follow the autoregressive chain rule and leaves residual discrepancies (see Sec. 3.2). In contrast, our Self-Rollout strategy strictly adheres to the chain rule and aligns training with inference, providing a more principled formulation for integration with reinforcement learning.

**Alignment for diffusion model.** Reinforcement learning (RL) post-training has demonstrated strong effectiveness in MLLMs (Li et al., 2025a; Peng et al., 2025; Wu et al., 2025) and has also been increasingly adopted in video generation. Existing approaches include scalar reward fine-tuning (Prabhudesai et al., 2023; Clark et al., 2023; Xu et al., 2023; Prabhudesai et al., 2024), Reward-Weighted Regression (RWR) (Peng et al., 2019; Lee et al., 2023; Furuta et al., 2024), and Direct Preference Optimization (DPO)-based methods (Rafailov et al., 2023; Wallace et al., 2024; Dong et al., 2023). However, policy gradient methods (Schulman et al., 2017; Fan et al., 2023) often suffer from instability. To improve stability in generative modeling, recent works such as Dance-GRPO (Xue et al., 2025) and FlowGRPO (Liu et al., 2025) extend GRPO to flow-matching models. Building on this line of research, we extend GRPO to the I2V setting, achieving improved motion controllability while maintaining visual quality and efficiency.

## 3 METHOD

Our `AR-Drag` has two steps: In step 1 (Section 3.2), we build a real-time AR base model with basic motion control ability—assemble control-aware data, train a bidirectional teacher, and distill to a few-step causal student; during distillation we introduce Self-Rollout to align training with AR inference. In step 2 (Section 3.3), we treat AR video generation as an MDP and optimize with GRPO, designing selective stochastic sampling and a reward to improve realism and motion control.

### 3.1 PRELIMINARY

**Flow matching.** Given a prior $p_0(\mathbf{x})$ and target data distribution $p_1(\mathbf{x})$, flow matching constructs an interpolating distribution $p_t(\mathbf{x})$. The sample trajectory $\mathbf{x}_t$ follows the probability flow ODE:

$$\frac{\mathrm{d}\mathbf{x}_t}{\mathrm{d}t} = \mathbf{v}_\theta(\mathbf{x}_t, t), \quad \mathbf{x}_0 \sim p_0. \tag{1}$$

The training objective minimizes the squared error between the predicted vector field $\mathbf{v}_\theta$ and the ground-truth flow $\mathbf{v}$:

$$\mathcal{L}_{\text{FM}}(\theta) = \mathbb{E}_{t,\mathbf{x}_t}[\|\mathbf{v}_\theta(\mathbf{x}_t, t) - \mathbf{v}\|_2^2], \tag{2}$$

where the target velocity field is $\mathbf{v} = \mathbf{x}_1 - \mathbf{x}_0$.

**Flow-ODE to SDE.** In flow-based probability models, the forward process is deterministic and follows an ODE: $\mathrm{d}\mathbf{x}_t = \mathbf{v}_t\mathrm{d}t$. To introduce stochasticity while preserving the same marginal distributions, a reverse-time SDE formulation can be defined as:

$$\mathrm{d}\mathbf{x}_t = \left(\mathbf{v}_t(\mathbf{x}_t) - \tfrac{1}{2}\sigma_t^2 \nabla \log p_t(\mathbf{x}_t)\right)\mathrm{d}t + \sigma_t \, \mathrm{d}\mathbf{w}, \tag{3}$$

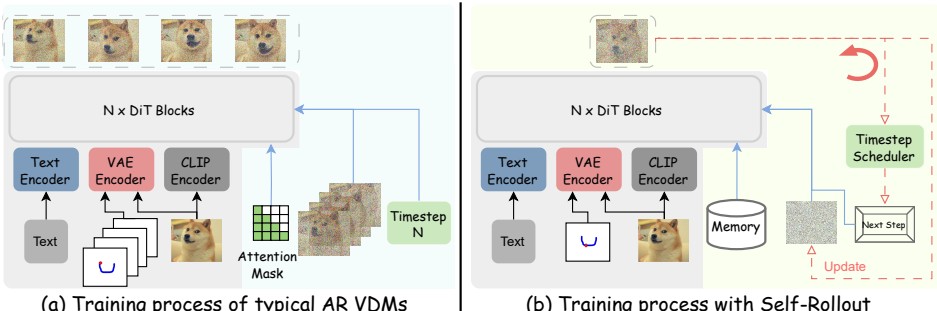

Figure 2: Comparison between typical AR VDMs and Self-Rollout. Self-Rollout faithfully follows the inference process during training, minimizing the train–test gap and naturally preserving the Markov property.

which leads to the update rule:

$$\mathbf{x}_{t+\Delta t} = \mathbf{x}_t + [\mathbf{v}_\theta(\mathbf{x}_t, t) + \tfrac{1}{2t}\sigma_t^2(\mathbf{x}_t + (1-t)\mathbf{v}_\theta(\mathbf{x}_t, t))]\Delta t + \sigma_t\sqrt{\Delta t}\epsilon. \tag{4}$$

**Distribution matching distillation (DMD).** DMD distills a multi-step teacher model into a few-step student model (Yin et al., 2024b;a) by minimizing the KL divergence between student-generated distribution $p_{\theta,t}$ and data distribution distribution $p_{\text{data},t}$ across randomly sampled time $t$:

$$\mathbb{E}_t\left[\text{KL}\left(p_{\theta,t}\|p_{\text{data},t}\right)\right] \tag{5}$$

## 3.2 STEP 1: FINE-TUNING A REAL-TIME MOTION-CONTROLLABLE BASE VDM

In Step 1, we build a base AR VDM with basic real-time motion control by (i) curating videos with control signals, (ii) fine-tuning a bidirectional VDM on this data to learn motion control, (iii) distilling it into a few-step causal AR model for real-time inference with Self-Rollout, which "Markovize" AR training and paves the way for GRPO in Step 2.

**Data curation.** We collect a training corpus of real and synthetic videos featuring diverse motions. Control signals are obtained by generating keypoint trajectories with an automatic detector (Doersch et al., 2022) and retaining only samples that pass human verification. For challenging cases, such as occlusion or fast motion, we additionally curate a high-quality dataset that is fully annotated by human annotators. Our curated corpus encompasses a rich spectrum of actions and visual styles—spanning humans, animals, and cartoons—and includes videos of varying resolutions and durations, making it well-suited for evaluating generalization across diverse scenarios. In addition, each video is accompanied by rich textual descriptions (both positive and negative prompts). Please refer to Appendix C.1 for the details.

**Bidirectional fine-tuning with motion-control.** At $m$-th frame, we use three control signals

$$\mathbf{c}_m = \begin{cases} \left(\mathbf{c}_m^{\text{traj}}, \mathbf{c}^{\text{text}}, \mathbf{c}^{\text{ref}}\right), & m = 0, \\ \left(\mathbf{c}_m^{\text{traj}}, \mathbf{c}^{\text{text}}, \varnothing\right), & \text{otherwise.} \end{cases} \tag{6}$$

Here, $\mathbf{c}_m^{\text{traj}}$ is a motion-trajectory embedding obtained by encoding the raw coordinate heatmap at frame $m$ with a VAE encoder (Wan et al., 2025). $\mathbf{c}^{\text{text}}$ encodes the textual signal, combining both positive and negative prompts. The text embedding is shared across all frames. At the initial frame ($m = 0$), the reference image embedding $\mathbf{c}^{\text{ref}}$ is encoded by a VAE encoder and a CLIP visual encoder (Radford et al., 2021). For subsequent frames ($m>0$), we do not condition on a reference image ($\varnothing$) and inject Gaussian noise in its place.

The model is trained with the flow matching objective, extended to incorporate control signals:

$$\mathcal{L}_{\text{FM}}(\theta) = \mathbb{E}_{t,\mathbf{x}_t}[\|\mathbf{v}_\theta(\mathbf{c}, t, \mathbf{x}_t) - \mathbf{v}\|_2^2], \tag{7}$$

where $\mathbf{c}$ denotes the full set of control inputs across the entire video, $e.g.$, $\mathbf{c} = \{\mathbf{c}_m\}_{m=0}^M$.

**Distilling to real-time AR model.** Following previous techniques (Huang et al., 2025; Yin et al., 2025), we distill the bidirectional teacher model into a few-step student model by replacing bidirectional attention with causal attention. The student is further optimized with DMD (Yin et al., 2024a) and adversarial losses (Goodfellow et al., 2020). Given a noise schedule $\mathcal{T} = \{t_0 = T, \ldots, t_N = 0\}$, each frame is denoised over $N$ steps, where $N$ is significantly smaller than that in multi-step VDMs, enabling real-time inference.

**Self-Rollout: Markovizing AR training.** Although an AR VDM conditions on its own generated history at inference, AR training typically uses teacher forcing—each step conditions on ground-truth past frames rather than model outputs—creating a train–test mismatch (exposure bias) and breaking the Markov property required for RL. As illustrated in Fig. 2 (a), noise is added to the ground-truth frame, and the model predicts the corresponding vector field.

To address this issue, we propose a Self-Rollout strategy, which maintains a key–value (KV) memory cache storing previously denoised frames as causal context. As shown in Fig. 2 (b), frames are denoised sequentially from pure noise during training. Let $\mathbf{x}_{m,n}$ denote the $m$-th frame at denoising step $n$. For the $m$-th frame, we randomly sample a denoising step $n$, denoise step-by-step from $\mathbf{x}_{m,0}$ to $\mathbf{x}_{m,n}$, and compute the DMD loss in Eq. (5) and adversarial loss. We then continue denoising from $\mathbf{x}_{m,n}$ to $\mathbf{x}_{m,N}$ step-by-step, updating the KV cache with the generated clean frame $\mathbf{x}_{m,N}$. In this way, subsequent frames are conditioned on the self-generated KV cache rather than ground-truth history. In contrast, Self-Forcing (Huang et al., 2025) updates the KV cache by collapsing the denoising trajectory from $\mathbf{x}_{m,n}$ to $\mathbf{x}_{m,N}$ into a single step. Our step-by-step Rollout more faithfully matches inference dynamics and naturally integrates with RL–based training.

### 3.3 STEP 2: REINFORCEMENT LEARNING ON AR VDM

Our Self-Rollout strategy (Sec. 3.2) "Markovizes" AR training by conditioning on model-generated histories and the ODE-to-SDE conversion in Eq. (4) supplies the stochasticity. Taken together, these resolve the two obstacles to applying GRPO—it requires an MDP and stochastic rollouts. In the sequel, we first set notations and formulate the MDP underlying video generation.

**Notations.** Consider a video of $M+1$ frames, each denoised in $N$ steps. We denote the $m$-th frame at denoising step $n$ by $\mathbf{x}_{m,n}$. Let $\mathbf{x}_{<m,N} = \{\mathbf{x}_{0,N}, ..., \mathbf{x}_{m-1,N}\}$ be the $m-1$ already denoised clean frames and $\mathbf{x}_{>m,0} = \{\mathbf{x}_{m+1,0}, ..., \mathbf{x}_{M,0}\}$ the unprocessed, noise-initialized frames. At state $(m, n)$, the video snapshot is

$$\mathbf{X}_{m,n} = \underbrace{\mathbf{x}_{<m,N}}_{\text{fully generated}} \cup \underbrace{\{\mathbf{x}_{m,n}\}}_{\text{being denoised}} \cup \underbrace{\mathbf{x}_{>m,0}}_{\text{initial noise}}, \tag{8}$$

The final clean video is then $\mathbf{X}_{M,N} = \{\mathbf{x}_{0,N}, \ldots, \mathbf{x}_{M,N}\}$. For autoregressive video generation, the denoising across frames produces a trajectory

$$\tau = \{\underbrace{\mathbf{X}_{0,0}, \mathbf{X}_{0,1}, \ldots, \mathbf{X}_{0,N}}_{\text{trajectory of frame 0}}, \underbrace{\mathbf{X}_{1,0}, \mathbf{X}_{1,1}, \ldots, \mathbf{X}_{1,N}}_{\text{trajectory of frame 1}}, \ldots, \underbrace{\mathbf{X}_{M,0}, \mathbf{X}_{M,1}, \ldots, \mathbf{X}_{M,N}}_{\text{trajectory of frame } M}\}. \tag{9}$$

**Video generation as MDP.** The denoising process in VDM can be formulated as a Markov decision process (MDP) (Liu et al., 2025; Xue et al., 2025):

- **State**: $\mathbf{s}_{m,n} \triangleq (\mathbf{c}_m, t_n, \mathbf{X}_{m,n})$, where $\mathbf{c}_m$ is the control signals. The initial-state distribution is $p(\mathbf{s}_{0,0}) = p(\mathbf{c}, t_0, \mathbf{X}_{0,0}) = p(\mathbf{c}_0)\, \delta(t - t_0) \prod_{m=0}^{M} \mathcal{N}(\mathbf{x}_{m,0} \mid \mathbf{0}, \mathbf{I})$, *i.e.*, the control $\mathbf{c}_0$ is drawn from its prior, $t$ is fixed to $t_0$, and all frames start from Gaussian. $\delta(\cdot)$ denotes the Dirac distribution.

- **Action**: $\mathbf{a}_{m,n} \triangleq \mathbf{x}_{m,n+1}$, *i.e.*, the next denoised state of the $m$-th frame at step $n+1$. The policy is parameterized by the VDM with $\theta$:

$$\mathbf{a}_{m,n} = \mathbf{x}_{m,n+1} \sim p_\theta(\cdot \mid \mathbf{c}_m, t_n, \mathbf{X}_{m,n}). \tag{10}$$

  where stochasticity is introduced through the ODE-to-SDE conversion in Eq. (4).

- **Transition**: *(1) intra-frame transition.* With in a frame, the transition is deterministic given the current state and action: $p(\mathbf{s} \mid \mathbf{s}_{m,n}, \mathbf{a}_{m,n}) = \delta(\mathbf{s} - \mathbf{s}_{m,n+1})$. *(2) inter-frame transition.* When denoising of frame $m$ is complete ($n = N$), the state transitions to the initial state of the next frame $m+1$:

$$\mathbf{s}_{m+1,0} = (\mathbf{c}_{m+1}, t_0, \mathbf{X}_{m+1,0}), \qquad \text{where} \quad \mathbf{X}_{m+1,0} = \mathbf{X}_{m,N} \quad \text{by definition.} \tag{11}$$

- **Reward function**: Rewards are provided only when a frame is fully denoised ($n = N$):

$$R(\mathbf{s}_{m,n}, \mathbf{a}_{m,n}) \triangleq R(\mathbf{x}_{m,N}, \mathbf{c}_m) = \mathbb{1}[n = N] \cdot (R_{\text{quality}}(\mathbf{x}_{m,N}) + R_{\text{motion}}(\mathbf{x}_{m,N}, \mathbf{c}_m)) \quad (12)$$

where $\mathbb{1}[\cdot]$ is the indicator function, $R_{\text{quality}}$ measures perceptual fidelity and temporal smoothness and $R_{\text{motion}}$ measures alignment with control signals. (We defer precise definitions to the sequel.)

**GRPO for AR VDM.** We extend GRPO framework to AR video generation. Under the MDP formulation, the AR VDM samples a group of $G$ videos $\{\mathbf{X}_{M,N}^{(i)}\}_{i=1}^{G}$ along with their trajectories $\{\tau^{(i)}\}_{i=1}^{G}$. The advantage of the $i$-th video is computed as:

$$\hat{A}_{m,n}^{(i)} = \frac{R(\mathbf{x}_{m,N}^{(i)}, \mathbf{c}_m) - \text{mean}(\{R(\mathbf{x}_{m,N}^{(j)}, \mathbf{c}_m)\}_{j=1}^{G})}{\text{std}(\{R(\mathbf{x}_{m,N}^{(j)}, \mathbf{c}_m)\}_{j=1}^{G})}. \quad (13)$$

The GRPO objective is defined as:

$$\mathcal{L}_{\text{GRPO}}(\pi_\theta) = \mathbb{E}_{\mathbf{c}, \{\tau^{(i)}\}_{i=1}^{G} \sim \pi_{\theta_{\text{old}}}(\cdot|\mathbf{c})}$$

$$\left[ \frac{1}{GMN} \sum_{i=1}^{G} \sum_{m=1}^{M} \sum_{n=1}^{N} \left( \min\left(r_{m,n}^{(i)}(\theta)\hat{A}_{m,n}^{(i)}, \text{clip}(r_{m,n}^{(i)}(\theta), 1 - \varepsilon, 1 + \varepsilon)\hat{A}_{m,n}^{(i)}\right) - \beta\text{KL}(\pi_\theta \| \pi_{\text{ref}}) \right) \right],$$
$$(14)$$

where the importance ratio is: $r_{m,n}^{(i)}(\theta) = p_\theta(\mathbf{x}_{m,n+1}^{(i)} \mid \mathbf{x}_{m,n}^{(i)}, \mathbf{c}_m) / p_{\theta_{\text{old}}}(\mathbf{x}_{m,n+1}^{(i)} \mid \mathbf{x}_{m,n}^{(i)}, \mathbf{c}_m)$.

**Selective stochastic sampling.** GRPO requires stochasticity for advantage estimation and policy exploration, which we introduce via the ODE-to-SDE conversion. However, in video generation the Markov chain is extremely long, and applying SDE sampling at every denoising step induces very high variance in trajectory returns, which substantially increases the number of rollouts ($G$) needed for stable loss estimation and thus incurs prohibitive cost.

To balance exploration and efficiency, we adopt *selective stochasticity*: a single denoising step $\tilde{n}$ is randomly chosen to follow the SDE formulation, while all remaining steps stay deterministic under the ODE solver. This strategy injects sufficient randomness for effective RL training, while maintaining computational efficiency.

**Reward design.** We design a composite reward that jointly evaluates visual realism ($R_{\text{quality}}$) and motion controllability ($R_{\text{motion}}$). For realism, we adopt the LAION Aesthetic Quality Predictor (Schuhmann, 2022) denoted as $f_{\text{AQ}}$ that assigns an aesthetic score (1-5) to each image. The realism reward is defined as

$$R_{\text{quality}}(\mathbf{x}_{m,N}) = f_{\text{AQ}}(\mathbf{x}_{m,N}). \quad (15)$$

For motion controllability, we employ Co-Tracker (Karaev et al., 2024) to first estimate the object trajectory $\hat{\mathbf{c}}_m^{\text{traj}}$ at frame $m$ from the generated image and measure their alignment with the ground-truth $\mathbf{c}_m^{\text{traj}}$. The motion reward is defined as

$$R_{\text{motion}}(\mathbf{x}_{m,N}, \mathbf{c}_m) = \lambda \max(0, \alpha - \|\hat{\mathbf{c}}_m^{\text{traj}} - \mathbf{c}_m^{\text{traj}}\|_2^2), \quad (16)$$

where $\alpha$ is an offset, and $\lambda$ is the scaling hyperparameter.

### 3.4 Discussion with Existing Techniques.

Our Self-Rollout eliminates this collapse entirely by continuing full step-by-step ancestral sampling using only the model's own predictions—identical to inference. Combined with selective stochasticity (Sec. 3.3), we reduce the effective horizon by 5–20× while preserving exploration, enabling stable and effective GRPO training on autoregressive video diffusion for the first time.

**Comparison with Self-Forcing** Although our Self-Rollout strategy and Self-Forcing (Huang et al., 2025) both address exposure bias by using self-generated context in autoregressive video diffusion, they differ fundamentally in how the KV cache is updated after the supervised prefix. These differences critically impact alignment with inference-time dynamics and compatibility with reinforcement learning objectives such as GRPO.

By performing a full step-by-step rollout instead of a single non-sequential collapse, Self-Rollout perfectly eliminates the train–inference distribution mismatch, provides a clean sequential decision

Table 1: Quantitative comparisons with motion-controllable VDMs. Best results are **bold**.

| Method | Latency (s) ↓ | FID ↓ | FVD ↓ | Aesthetic Quality ↑ | Motion Smoothness ↑ | Motion Consistency ↑ |
|---|---|---|---|---|---|---|
| DragNUWA | 94.26 | 36.31 | 376.39 | 3.30 | 0.9759 | 3.71 |
| DragAnything | 68.76 | 38.13 | 367.74 | 3.22 | 0.9811 | 3.63 |
| Tora | 176.51 | 32.84 | 283.43 | 3.86 | 0.9855 | 3.97 |
| MagicMotion | 1426.37 | 30.04 | 230.53 | 4.01 | 0.9871 | 3.95 |
| Self-Forcing | 0.95 | 34.47 | 315.87 | 3.70 | 0.9920 | 4.06 |
| AR-Drag | **0.44** | **28.98** | **187.49** | **4.07** | **0.9948** | **4.37** |

process that GRPO can directly optimize, and—when combined with selective stochasticity sampling—effectively mitigates the extremely long-horizon problem. This enables successful application of GRPO to high-fidelity autoregressive video generation for the first time.

## 4 EXPERIMENTS

**Implementation details.** We implement our base model with Wan2.1-1.3B-I2V (Wan et al., 2025), using a 3-step diffusion process in a frame-wise manner, denosing one latent at a time. To accommodate varying resolutions, we define a set of bucket sizes and resize each video to its nearest bucket. The KV cache is set to hold 7 frames; when updating the cache, the oldest frame is removed if the cache exceeds this size. All training is performed using the AdamW optimizer (Loshchilov & Hutter, 2017) with a learning rate of $1 \times 10^{-5}$, on 8 NVIDIA H20 GPUs. We do not adopt LoRA fine-tuning, as it may introduce long-tail performance degradation (Tian et al., 2024; 2025; Song et al., 2024). For evaluation, we curate a new benchmark consisting of 206 video clips covering diverse motion trajectories and scene variations, specifically designed to assess motion controllability.

**Metrics.** We adopt standard metrics such as Fréchet Inception Distance (FID) (Seitzer, 2020), Fréchet Video Distance (FVD) (Unterthiner et al., 2018), and Aesthetic Quality (Schuhmann, 2022) to quantitatively evaluate visual quality. To assess motion controllability, we employ two complementary measures: Motion Smoothness (Huang et al., 2024), which captures the stability of motion across frames, and Motion Consistency, which evaluates the alignment between control trajectories and the resulting motion dynamics, computed using our proposed reward model. We report first-frame latency calculated on a single NVIDIA H20 GPU as an indicator of real-time performance.

**Baselines.** We compare our method against strong open-source motion-guided VDMs, including DragNUWA (Yin et al., 2023), DragAnything (Wu et al., 2024), Tora (Zhang et al., 2025) and Magicmotion (Li et al., 2025b). Following prior work (Zhang et al., 2025), we improve DragNUWA by adoping its motion trajectory design to a DiT-based architecture. Tora is the first one to apply DiT in this task, and MagicMotion further support complex trajectories-based controls. Since no AR motion-control I2V baseline is available, we fine-tune a chunk-wise AR VDM, Self-Forcing (Huang et al., 2025), which was originally designed for text-to-video (T2V) generation. Specifically, we fine-tune Wan2.1-1.3B-I2V following the Self-Forcing architecture and training procedure using the same datasets as AR-Drag. In this adaptation, the model denoises three latents simultaneously in each denoising loop to achieve effective motion controllability.

### 4.1 RESULTS

**Quantitative comparisons.** The overall performance comparisons are reported in Tab. 1, leading to the following key observations: Our method **significantly reduces latency**. It requires only 0.44s, while bidirectional approaches such as Tora take 176.51s—less than 1% of their latency. For the 5B model MagicMotion, the latency is even higher at 1426.37 s. Thanks to the few-step distillation and causal design, our model can produce results immediately once the first frame is generated..

Despite being a few-step autoregressive design, AR-Drag still delivers **the best visual quality**. Specifically, it achieves the lowest FID and FVD, as well as the highest Aesthetic Quality, reflecting superior visual fidelity and temporal coherence. In terms of motion control metrics, our model attains the highest motion smoothness and consistency, highlighting its strength in precise and stable motion control. This contributes to our RL post training, which incentivizes the model's

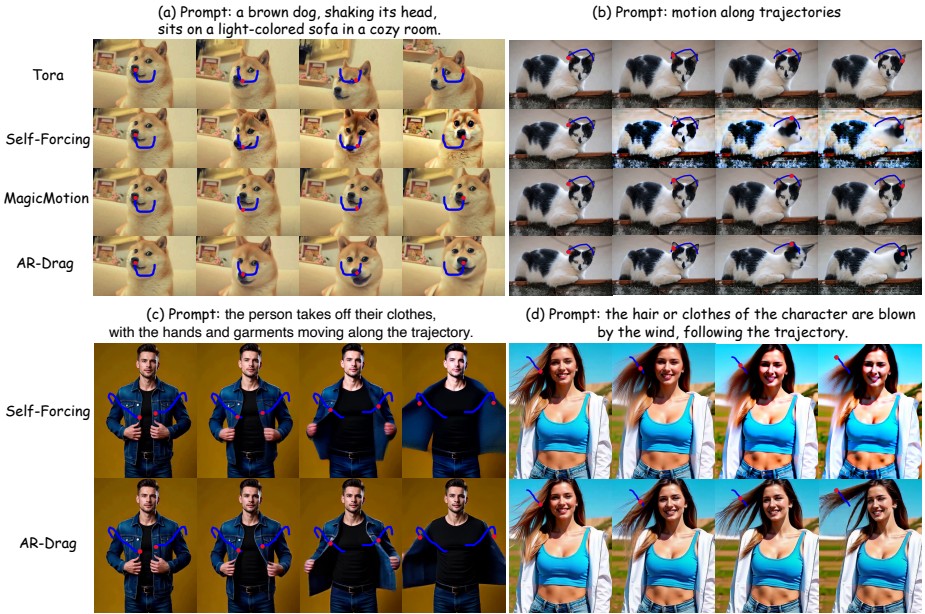

Figure 3: Qualitative comparisons with Tora and Self-Forcing across different prompts, data domains, and resolutions, demonstrating the superior fidelity and controllability of our method.

ability to follow motion guidance, enabling more flexible and robust controllability. Remarkably, `AR-Drag` even outperforms the 5B MagicMotion, particularly on motion-control. MagicMotion does not utilize RL training, which limits its ability to achieve fine-grained, highly flexible control.

Self-Forcing baseline also adopts a few-step AR design, but requires 0.95s—more than twice our latency—since it denoises three frames simultaneously. Moreover, `AR-Drag` outperforms Self-Forcing in both visual quality and motion control. These results demonstrate the effectiveness of our RL post-training and Self-Rollout for real-time motion-controllable video generation.

**Qualitative comparisons.** We conduct qualitative comparisons with three competitive baselines, Tora, MagicMotion and Self-Forcing. As shown in Fig. 3, we evaluate across different prompts, ranging from specific actions such as head shaking and taking off clothes, to more general motions such as following a trajectory. We further compare performance on both synthetic data (a), (c), (d) and real-world data (b), as well as across different resolutions. Since Tora only supports a fixed resolution, the resolution-based comparison in (c) and (d) is conducted only against Self-Forcing. For clarity, we visualize the entire trajectory across frames in blue and highlight the control signal of the current frame in red. The reference image is provided for the first frame. Since the same negative prompt is applied to all videos, only the positive prompt is shown.

As illustrated in Fig. 3(a&b), Tora and MagicMotion struggle to maintain consistency with the control signals. Self-Forcing achieves partial controllability but suffers from deformation and quality degradation. In contrast, our method delivers superior fidelity and control alignment. Furthermore, as shown in Fig. 3(c&d), Self-Forcing exhibits substantial detail loss—particularly in fine structures such as fingers and hair strands—and suffers from increased color saturation in (c), whereas our method consistently preserves high-quality details and maintains faithful motion control.

## 4.2 ABLATION STUDIES

In Tab. 2, we present the ablations on key training strategies.

**w/o RL.** Removing reinforcement learning leads to a noticeable drop in both quality and motion-related metrics, highlighting the importance of RL in enhancing fidelity and motion controllability.

Table 2: Ablation study on key training strategies. 'w/o RL' denotes removing the RL post-training. 'Initial model' refers to Wan2.1-1.3B-I2V prior to adaptation. 'Teacher model' is the fine-tuned multi-step bidirectional model. 'w/o Self-Rollout' denotes training without the Self-Rollout design.

| Method | Latency (s) ↓ | FID ↓ | FVD ↓ | Aesthetic Quality ↑ | Motion Smoothness ↑ | Motion Consistency ↑ |
|---|---|---|---|---|---|---|
| AR-Drag | 0.44 | 28.98 | 187.49 | 4.07 | 0.9948 | 4.37 |
| w/o RL | 0.44 | 31.65 | 210.35 | 3.92 | 0.9926 | 4.12 |
| Initial model | 45.72 | 35.94 | 303.16 | 3.84 | 0.9915 | 3.22 |
| Teacher model | 45.64 | 29.38 | 151.46 | 4.15 | 0.9941 | 4.36 |
| w/o Self-Rollout | 0.44 | 38.13 | 353.75 | 3.38 | 0.9904 | 4.02 |

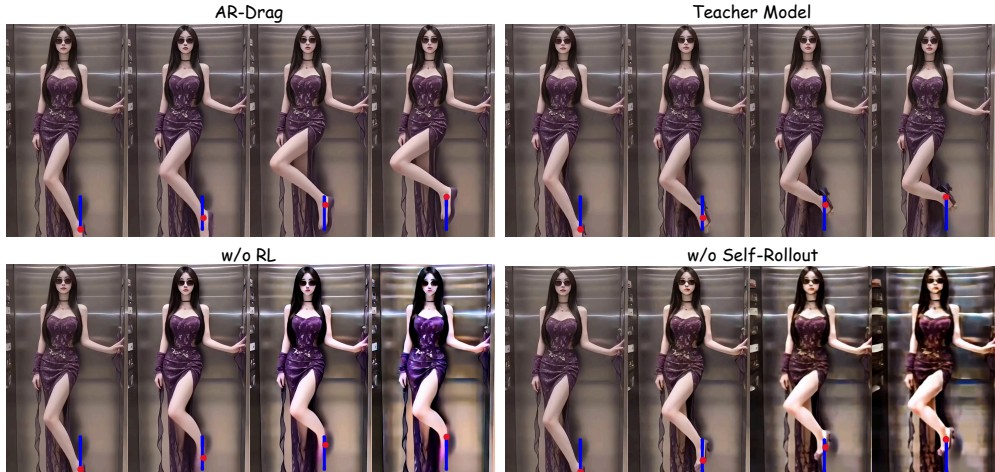

Figure 4: Ablation on key training strategies. Prompt: movement following the trajectory.

**Initial model.** The initial Wan2.1-1.3B-I2V model performs worse than our base model (w/o RL) on video quality and have a high latency, demonstrating that our motion fine-tuning and real-time post-training strategies provide a strong foundation for RL training.

**Teacher model.** The teacher model, a fine-tuned bidirectional multi-step baseline, achieves strong performance but suffers from high latency. While it represents the upper bound of DMD-based method, our AR-Drag achieves comparable or even better results in FID, Aesthetic Quality, Motion Smoothness, and Motion Consistency, confirming the effectiveness of our RL approach.

**w/o Self-Rollout.** Removing the Self-Rollout design leads to severe quality degradation, underscoring its necessity for maintaining the Markov property and mitigating the train-test mismatch in autoregressive generation.

**Visualization.** Since the initial model performs significantly worse, we exclude it from the comparison. As shown in Fig. 4, due to the absence of the feet in the reference image, both the teacher model and the model without RL fail to generate clear foot details, reflecting limited generalization. In contrast, our RL-based method encourages exploration, enhancing the model's generalization capability. Additionally, the model w/o RL exhibits increased color saturation, while the model without Self-Rollout suffers from severe image artifacts and quality degradation, caused by the train–test discrepancy and the disruption of the Markov property.

**Visualization on diverse motion.** We show qualitative results of our model conditioned on different motion trajectories in Fig. 5. The results demonstrate that our method can accurately follow diverse motion commands, while preserving visual quality, and temporal consistency across frames.

## 5 DISCUSSION

**Difference between Self-Rollout and Self-Forcing.** Applying GRPO to AR VDMs requires the base model to follow the Markov Decision Chain. However, standard AR VDMs exhibit a train-

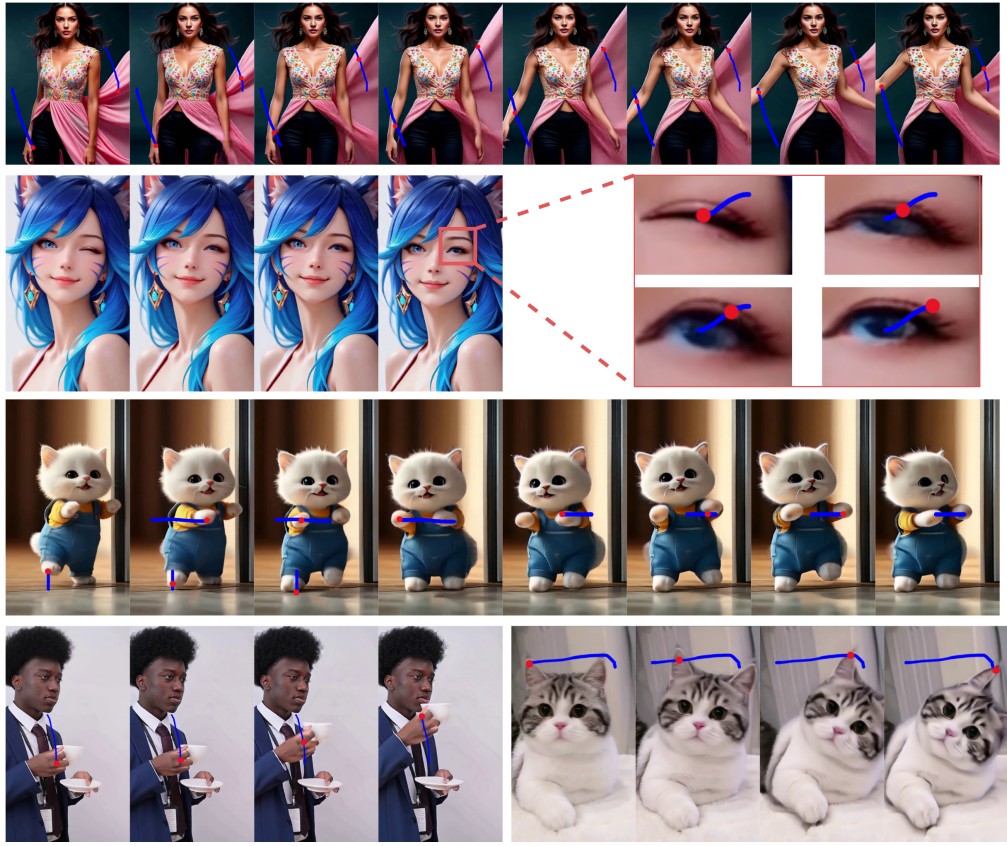

Figure 5: Visualization on diverse motion. Prompt: movement following the trajectory.

ing–inference gap—training conditions on ground truth while inference conditions on generated frames—breaking the Markov property and preventing direct GRPO training. Self-Forcing partially reduces this gap by using self-generated frames as KV cache (Alg. 2), but it skips remaining denoising steps and thus still violates the MDP. Our Self-Rollout enforces the full denoising rollout for every frame, restoring the proper Markov structure. This simple but critical correction makes RL training valid and leads to clear performance gains, as shown in Table 1, Figure 3, and the 'w/o Self-Rollout' ablations in Table 2 and Figure 4. More details can be found in Appendix A.

**Importance of selective stochastic sampling.** In bidirectional VDMs, frames are denoised jointly, so the decision-chain length equals the number of denoising steps. In AR VDMs, however, frames are denoised sequentially, making the chain length scale with denoising steps × frame count, leading to extremely long horizons. This causes return variance to explode and gradient estimates to become unusably noisy, making direct GRPO (or any policy-gradient method) practically infeasible. Our selective stochasticity sampling provides controlled exploration at each step without triggering variance explosion, enabling stable and sample-efficient GRPO training for AR video generation.

## 6 CONCLUSION

We present `AR-Drag`, the first RL-enhanced few-step autoregressive video diffusion model for real-time motion-controllable image-to-video generation. By combining selective stochasticity, and a trajectory-based reward model, our approach effectively addresses the challenges of quality degradation, motion artifacts, and complex control spaces in few-step AR video generation. Extensive experiments demonstrate that `AR-Drag` achieves high visual fidelity, precise motion alignment, and significantly lower latency compared with state-of-the-art motion-controllable VDMs, while maintaining a compact model size of only 1.3B parameters.

## ACKNOWLEDGEMENTS

This research is supported by the National Research Foundation, Singapore under its AI Singapore Programme (AISG Award No: AISG3-RP-2022-030). This project is also partially supported by the Ministry of Education, Singapore, under its Tier-1 Academic Research Fund (No. 24-SIS-SMU-040). This research is also supported by National Research Foundation, Singapore, NRF-NRFI10-2024-0004.

## ETHICS STATEMENT

This work presents a method for real-time controllable video generation. Our experiments are conducted on de-identified datasets that do not contain personally identifiable information. The study is intended solely for scientific research, and we adhere to the ICLR Code of Ethics regarding fairness, integrity, and responsible use of data and models.

## REPRODUCIBILITY STATEMENT

We provide detailed implementation settings, including model architecture, training objectives, optimization strategies, and hyperparameters in the main text and Appendix. The code, configuration files, and instructions for reproducing the main experiments are available in Supplementary Materials to facilitate verification and further research.

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

## A  LIMITATION AND FUTURE WORK

Our generative model is trained on data that follows physical plausibility, and our reward model is also designed to evaluate motion based on physical principles. Therefore, if a user intentionally provides highly exaggerated or physically impossible control signals, the model may not strictly follow such inputs because they fall outside the distribution it is trained and rewarded to respect. Handling deliberately non-physical or cartoon-like motion is an interesting direction for future work, and we believe extending controllability beyond physically plausible dynamics is a valuable avenue for exploration. Although our method already achieves real-time motion-controllable video generation, future work could further reduce wall-clock latency by trading additional parallel compute for faster inference, e.g., via parallel sampling (Zhu et al., 2025a; Wang et al., 2025c).

## B  MORE DETAILS ABOUT ARCHITECTURE

To better illustrates the difference between Self-Rollout and Self-Forcing, we provide the detailed training process in Alg. 1 and Alg. 2, respectively. As shown in Line 7 of Alg. 2, Self-Forcing randomly selects $t_s$ from denoising schedule and use the output at this step to compute loss (Line 9), ensuring that all denoising steps could be optimized. However, it directly treats the output at this intermediate denoising step as the final generated clean frame (Line 8), and uses it to update the KV cache (Line 10). This skips the remaining denoising steps from s to N, which breaks the MDP defined in Eq. 9. And the incorrect KV cache subsequently affects future generation. To address this issue, our Self-Rollout continues the denoising process all the way to step $N$, enforcing the full MDP transition defined in Eq. 9. This complete rollout is implemented in Lines 18–28 of Alg. 1.

In addition, we also provide the pseudo-code of Self-Rollout and Self-Forcing in Listing 1 and 2.

## C  MORE EXPERIMENTAL SETTINGS

### C.1  DATA CURATION

We construct our training corpus by combining both real and synthetic videos to cover diverse motion patterns. For the real videos, we directly collect footage from real-world sources. For the synthetic videos, we use Wan2.1-14B-I2V to generate videos containing a wide range of motion types but without control signals. In total, we gather approximately 10,000 videos. We then generate trajectory-based control signals using an automatic detector, followed by manual filtering to remove videos containing sensitive content, low-quality samples, or incorrect trajectories. For challenging scenarios, such as severe occlusion or fast motion, we curate a high-quality subset of approximately 3,000 videos, all of which are fully annotated by human annotators. Control signals include motion trajectories, prompts, and reference images. For motion trajectories, to better simulate actual user interactions, we represent each point as a bright spot with intensity ranging from 0 to 1 rather than a single isolated coordinate, mimicking the user's touch force on each frame.

For prompts, we provide both negative and positive prompts. The negative prompt is shared across all videos and follows the template:

> ### Negative Prompt Template
>
> *Overly vivid colors, overexposed, static, blurry details, subtitles, style, artwork, frame, still, overall grayish, worst quality, low quality, JPEG compression artifacts, ugly, incomplete, extra fingers, poorly drawn hands, poorly drawn faces, deformed, disfigured, malformed limbs, fused fingers, motionless frame, cluttered background, three legs, many people in the background, walking upside down.*

For positive prompts, we include either general motions along trajectories or specific actions to guide the desired video content.

To handle videos of varying resolutions, we define a set of predefined "bucket sizes" and resize each input video to its nearest bucket. The buckets include resolutions such as 480×368, 400×400,

---

**Algorithm 1** Self-Rollout Training

---

**Require:** Denoising schedule $\{t_0, t_1, \ldots, t_N\}$, Number of frames $M + 1$, model $G_\theta$, $\{\mathbf{c}_0, \mathbf{c}_1, \ldots, \mathbf{c}_N\}$
 1: **loop**
 2:     Initialize model output $\mathbf{X}_\theta \leftarrow [], KV cache$ KV $\leftarrow []$
 3:     Sample $s \sim \text{Unif}\{1, \ldots, N\}$
 4:     **for** $m = 0$ **to** $M$ **do**
 5:         Initialize $\mathbf{x}_{m,0} \sim \mathcal{N}(\mathbf{0}, I)$
 6:         **for** $n = 0$ **to** $s$ **do**
 7:             **if** $n = s$ **then**                 ▷ Ensure all denoising steps could be optimized
 8:                 `Enable gradient computation`
 9:                 $\hat{\mathbf{x}}_{m,N} \leftarrow G_\theta(\mathbf{x}_{m,s}, t_s, \mathbf{c}_m, \text{KV})$
10:                 $\mathbf{X}_\theta.\text{append}(\hat{\mathbf{x}}_{m,N})$
11:             **else**
12:                 `Disable gradient computation`
13:                 $\hat{\mathbf{x}}_{m,N} \leftarrow G_\theta(\mathbf{x}_{m,n}, t_n, \mathbf{c}_m, \text{KV})$
14:                 Sample $\epsilon \sim \mathcal{N}(\mathbf{0}, I)$
15:                 $\hat{\mathbf{x}}_{m,k-1} \leftarrow \Psi(\hat{\mathbf{x}}_{m,N}, \epsilon, t_{k-1})$
16:             **end if**
17:         **end for**
18:         $\hat{\mathbf{x}}_{m,s+1} \leftarrow \Psi(\hat{\mathbf{x}}_{m,N}, \epsilon, t_{s+1})$
19:         **for** $n = s + 1$ **to** $N$ **do**
20:             **if** $n = s$ **then**                 ▷ Enforce the MDP defined in Eq. 9
21:                 $\hat{\mathbf{x}}_{m,N} \leftarrow G_\theta(\mathbf{x}_{m,s}, t_s, \mathbf{c}_m, \text{KV})$
22:                 $\text{kv}_m \leftarrow G_\theta(\hat{\mathbf{x}}_{m,N}, \text{KV})$       ▷ Update KV cache with the right generation
23:                 KV.append($\text{kv}_m$)
24:             **else**
25:                 $\hat{\mathbf{x}}_{m,N} \leftarrow G_\theta(\mathbf{x}_{m,n}, t_n, \mathbf{c}_m, \text{KV})$
26:                 Sample $\epsilon \sim \mathcal{N}(\mathbf{0}, I)$
27:                 $\hat{\mathbf{x}}_{m,n+1} \leftarrow \Psi(\hat{\mathbf{x}}_{m,N}, \epsilon, t_{n+1})$
28:             **end if**
29:         **end for**
30:     **end for**
31:     Update $\theta$ on $\mathbf{X}_\theta$
32: **end loop**

---

368×480, 640×368, and 368×640. This strategy ensures consistent input dimensions while preserving aspect ratios as much as possible.

Table 3: Parameter analysis.

| Method | | Latency (s) ↓ | FID ↓ | FVD ↓ | Aesthetic Quality ↑ | Motion Smoothness ↑ | Motion Consistency ↑ |
|---|---|---|---|---|---|---|---|
| AR-Drag | | 0.44 | 28.98 | 187.49 | 4.07 | 0.9948 | 4.37 |
| chunk size | 3 | 0.94 | 27.47 | 179.23 | 4.09 | 0.9945 | 4.37 |
| cache size | 15 | 0.44 | 28.96 | 188.08 | 4.07 | 0.9946 | 4.34 |
| | 25 | 0.46 | 28.99 | 185.31 | 4.05 | 0.9948 | 4.39 |

## C.2 IMPLEMENTATION DETAILS

We implement our base model using Wan2.1-1.3B-I2V (Wan et al., 2025), employing a 3-step diffusion process with $N = 3$, and timesteps $t_0 = 1000, t_1 = 755, t_2 = 522, t_3 = 0$. We set chunk size as 1, cache size as 7. For distillation post training, we set DMD loss weight as 1, generator loss weight as 0.1, discriminator loss as 0.05.

---

**Algorithm 2** Self-Forcing Training

---

**Require:** Denoising schedule $\{t_0, t_1, \ldots, t_N\}$, Number of frames $M+1$, model $G_\theta$, control signals $\{\mathbf{c}_0, \mathbf{c}_1, \ldots, \mathbf{c}_N\}$

1: **loop**
2:     Initialize model output $\mathbf{X}_\theta \leftarrow [], KVcache\ \text{KV} \leftarrow []$
3:     Sample $s \sim \text{Unif}\{1, \ldots, N\}$
4:     **for** $m = 0$ **to** $M$ **do**
5:         Initialize $\mathbf{x}_{m,0} \sim \mathcal{N}(\mathbf{0}, I)$
6:         **for** $n = 0$ **to** $s$ **do**
7:             **if** $n = s$ **then**                ▷ One-step collapse from s to N, which skip steps in MDP
8:                $\hat{\mathbf{x}}_{m,N} \leftarrow G_\theta(\mathbf{x}_{m,s}, t_s, \mathbf{c}_m, \text{KV})$
9:                $\mathbf{X}_\theta.\text{append}(\hat{\mathbf{x}}_{m,N})$
10:              $\text{kv}_m \leftarrow G_\theta(\hat{\mathbf{x}}_{m,N}, \text{KV})$         ▷ Update KV cache with the collapsed generation
11:              $\text{KV}.\text{append}(\text{kv}_m)$
12:             **else**
13:                $\hat{\mathbf{x}}_{m,N} \leftarrow G_\theta(\mathbf{x}_{m,n}, t_n, \mathbf{c}_m, \text{KV})$
14:                Sample $\epsilon \sim \mathcal{N}(\mathbf{0}, I)$
15:                $\hat{\mathbf{x}}_{m,n+1} \leftarrow \Psi(\hat{\mathbf{x}}_{m,N}, \epsilon, t_{n+1})$
16:             **end if**
17:         **end for**
18:     **end for**
19:     Update $\theta$ on $\mathbf{X}_\theta$
20: **end loop**

---

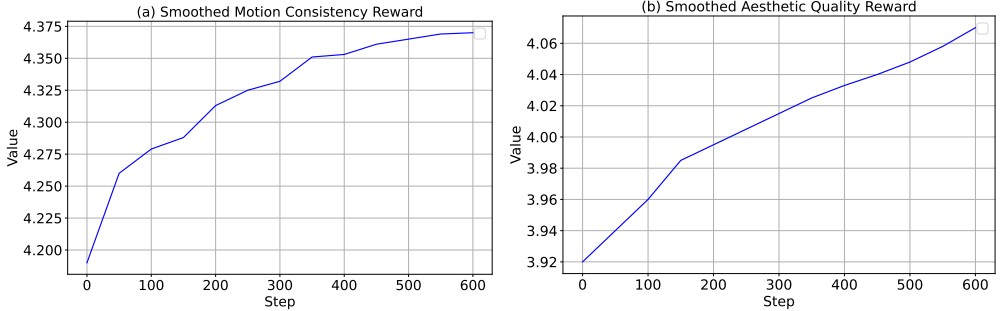

Figure 6: Smoothed Reward Curves for Motion Consistency and Aesthetic Quality

# D   MORE ANALYSIS

## D.1   PARAMETER ANALYSIS

We conduct parameter analysis, as shown in Tab. 3.

**Chunk size.** Typical AR VDMs operate in a chunk-wise manner, applying bidirectional attention within each chunk and causal attention across chunks. During inference, chunk-wise AR VDMs denoise all frames in a chunk simultaneously, which introduces some latency. In contrast, our approach adopts a frame-wise strategy, denoising one latent at a time. While this increases the potential for error accumulation, the combination of Self-Rollout and RL post-training allows us to achieve comparable performance even with a chunk size of 3.

**Cache size.** We set the KV cache size to 7 in our experiments. During inference, when the cache exceeds this length, the earliest frame is removed to maintain the fixed size. We observe that varying the cache size has little impact on the final performance, indicating that our method is robust to different cache lengths.

```python
def self_rollout_training(model, x_gt, cond_list, schedule):
    # schedule = [t_0, t_1, ..., t_N], len(schedule) = N+1
    # cond_list = [c_0, c_1, ..., c_M]
    M = len(cond_list) - 1
    X_theta = []            # collect supervised clean predictions
    KV = []                 # KV cache

    # Sample random supervised prefix length
    s = random.randint(0, N)           # Unif{0, ..., N}

    for m in range(M+1):
        c_m = cond_list[m]
        x = torch.randn_like(x_gt[m])    # x_{m,0} ~ N(0,I)

        # ===Phase 1: Supervised prefix (0 to s) ===
        for n in range(s + 1):           # n = 0,1,...,s
            if n == s:
                # Last supervised step: gradient flows
                torch.enable_grad()
                # predict clean frame
                x_hat_N = model(x, schedule[n], c_m, KV)
                X_theta.append(x_hat_N)
            else:
                torch.no_grad()
                x_hat_N = model(x, schedule[n], c_m, KV)
                epsilon = torch.randn_like(x_hat_N)
                x = reverse_step(x_hat_N, epsilon, schedule[n+1])
        x = reverse_step(x_hat_N, epsilon, schedule[s+1])
        # ===Phase 2: Self-generated rollout (s+1 to N) ===
        torch.no_grad()
        for n in range(s + 1, N + 1):
            # Final step: update KV cache
            if n == N:
                x_hat_N = model(x, schedule[n], c_m, KV)
                # extract KV from clean frame
                kv_m = model.get_kv(x_hat_N, KV)
                KV.append(kv_m)
            else:
                x_hat_N = model(x, schedule[n], c_m, KV)
                epsilon = torch.randn_like(x_hat_N)
                x = reverse_step(x_hat_N, epsilon, schedule[n+1])

    # Update model using DMD loss on collected clean frames
    loss = loss_func(X_theta, x_gt)
    loss.backward()
    optimizer.step()
    optimizer.zero_grad()
```

Listing 1: Pseudo code for Self-Rollout

## D.2 VISUALIZATION OF REWARD CURVES.

Fig. 6 illustrates the training dynamics of our two reward signals: Smoothed Motion Consistency Reward and Smoothed Aesthetic Quality Reward. Both curves show a clear upward trend as training progresses, reflecting the model's improving ability to maintain coherent motion and generate visually appealing outputs. The motion consistency reward rises steadily, indicating better alignment with the target trajectories, while the aesthetic reward demonstrates rapid gains in the early stages followed by a slower convergence, suggesting progressive refinement in visual quality. Together, these smoothed reward curves highlight the effectiveness of our reinforcement learning design in balancing motion control and perceptual quality.

```python
1  def self_rollout_training(model, x_gt, cond_list, schedule):
2      # schedule = [t_0, t_1, ..., t_N], len(schedule) = N+1
3      # cond_list = [c_0, c_1, ..., c_M]
4      M = len(cond_list) - 1
5      X_theta = []            # collect supervised clean predictions
6      KV = []                 # KV cache
7
8      # Sample random supervised prefix length
9      s = random.randint(0, N)            # Unif{0, ..., N}
10
11     for m in range(M+1):
12         c_m = cond_list[m]
13         x = torch.randn_like(x_gt[m])    # x_{m,0} ~ N(0,I)
14
15         # ===Phase 1: Supervised prefix (0 to s) ===
16         for n in range(s + 1):           # n = 0,1,...,s
17             if n == s:
18                 # Last supervised step: gradient flows
19                 torch.enable_grad()
20                 # predict clean frame
21                 x_hat_N = model(x, schedule[n], c_m, KV)
22                 X_theta.append(x_hat_N)
23                 # extract KV from collapsed generation
24                 kv_m = model.get_kv(x_hat_N, KV)
25                 KV.append(kv_m)
26             else:
27                 torch.no_grad()
28                 x_hat_N = model(x, schedule[n], c_m, KV)
29                 epsilon = torch.randn_like(x_hat_N)
30                 x = reverse_step(x_hat_N, epsilon, schedule[n+1])
31         x = reverse_step(x_hat_N, epsilon, schedule[s+1])
32
33
34     # Update model using DMD loss on collected clean frames
35     loss = loss_func(X_theta, x_gt)
36     loss.backward()
37     optimizer.step()
38     optimizer.zero_grad()
```

Listing 2: Pseudo code for Self-Forcing.

## E  LLM USAGE STATEMENT

ChatGPT was employed solely for minor editorial assistance, such as improving grammar and readability. The research ideas, methodology, experiments, and analysis were entirely developed and conducted by the authors without the use of LLMs.

