# OpenReview forum: "Real-Time Motion-Controllable Autoregressive Video Diffusion"
_ICLR.cc/2026/Conference — ICLR 2026 Poster_

### Official Review · Reviewer_RqLj · 2025-10-26

**Soundness:** 3
**Presentation:** 2
**Contribution:** 3
**Rating:** 6
**Confidence:** 3

**Summary:**

This paper proposes AR-Drag, a trajectory-based real-time video diffusion model. AR-Drag is built on top of WAN-2.1-1.3B, using motion keypoints to define the motion trajectory. To further improve motion quality and latencies, the paper introduces two designs: i) distilling bi-directional attention in the base model into a few-step autoregressive model with self-rollout, similar to self-forcing but applying DMD on the randomly sampled n-th denoised frame; ii) the model generates history frames, leading to a stochastic MDP rollout that optimises GRPO with motion controllability and visual realism rewards. These improvements result in significantly reduced latencies and enhanced motion and generation quality compared to previous methods.

**Strengths:**

The paper is well-written and has a clean, straightforward design. The modification of self-forcing to introduce selective stochasticity on the sampled demonised step n is a smart design.  The inclusion of the source code is highly appreciated.

**Weaknesses:**

- The attached website is rushed and incomplete. It lacks examples of generations from the baselines and without visualisation of the conditioning motion trajectories, and it includes wrong input prompts. This makes it difficult for the reviewer to appreciate the generation qualities, especially for a videogen model. These visualisations would be crucial for supporting the experiment.
- Directly optimising test metrics within model training is slightly cheating compared to other baselines. While I understand GRPO is an essential part of the design, I feel like highlighting FID/FVD alone would be more fair. Additionally, having these metrics purely as an ablation for the RL component, as shown in Table 2, where the performance is also very good, seems sufficient.
- The training data collected by this paper is also important. The authors should provide more details about the collected data, including the source of the videos, how they are filtered, and the total data size of the videos. I am wondering whether the authors also consider open-sourcing the collected data?
- Considering the core design of the self-rollout is very similar to self-forcing, I would suggest adding a section on a side-by-side comparison of the implementation levels (like pseudo-code) to highlight the implementation differences. This would provide a clearer understanding of the method design.

**Questions:**

See the weaknesses.

---

> ### Author Response · Authors · 2025-11-23
> **Feedback to Reviewer #RqLj**
>
> We appreciate your overall positive assessment of our contributions and are grateful for your suggestion. We have already updated the new revision based on your suggestions.
>
> ------
>
> **W1. The attached website is rushed and incomplete. It lacks examples of generations from the baselines and without visualisation of the conditioning motion trajectories, and it includes wrong input prompts.**
>
> **A1**: Thanks for your suggestion. We have added more visualization results on our project page and also included generations from competitive baselines for comparison. To make the controllability clearer, we now provide the corresponding motion trajectories for every example.
>
> Our project page is organized into three sections.
> Section 1 compares AR-Drag with Tora, MagicMotion, and Self-Forcing. Tora and MagicMotion fail under fine-grained, highly flexible control, while Self-Forcing exhibits severe quality degradation.
> Section 2 compares AR-Drag with Self-Forcing, the pretrained backbone, and the fine-tuned teacher model. Self-Forcing again shows strong degradation, and the pretrained backbone struggles with motion consistency. AR-Drag achieves **quality comparable to the fine-tuned teacher model**—which serves as an approximate upper bound—and, thanks to RL post-training, even surpasses it in motion consistency.
> Section 3 demonstrates streaming generation with multiple control inputs. AR-Drag naturally supports streaming input, enabling better temporal consistency and higher visual quality. Bidirectional models (the pretrained backbone and the teacher model) cannot perform streaming generation, so for comparison we feed all control signals jointly when comparing against them.
>
> We also apologize for the incorrect input prompts in the earlier version. They have been fixed, and we hope this will avoid any confusion.
>
> ------
>
> **W2. Directly optimising test metrics within model training is slightly cheating compared to other baselines. While I understand GRPO is an essential part of the design, I feel like highlighting FID/FVD alone would be more fair. Additionally, having these metrics purely as an ablation for the RL component, as shown in Table 2, where the performance is also very good, seems sufficient.**
>
> **A2**: Thanks for your question. We use Aesthetic Quality and Motion Consistency as our optimized objection. For other metrics, FID, FVD, and Motion Smoothness, our AR-Drag also outperforms other baselines. In addition, to ensure that our evaluation remains valid and robust, we complement quantitative metrics with qualitative analysis. As shown in Figure 3, our method achieves consistently better visual quality and motion alignment compared with all baselines. In other words, AR-Drag outperforms competing methods both in FID/FVD scores and in actual perceptual quality, confirming that our improvements are not due to reward hacking.
>
> For FID/FVD, our RL component also brings substantial improvements. In addition, the qualitative analysis in Figure 4 further demonstrates that RL enhances the base model’s generalization ability. When the reference image lacks visible feet, the base model without RL fails to generate clear foot details, indicating limited generalization capability. In contrast, our RL-trained model infers the missing structure more accurately.

---

> ### Author Response · Authors · 2025-11-23
> **Feedback to Reviewer #RqLj**
>
> **W3. The authors should provide more details about the collected data, including the source of the videos, how they are filtered, and the total data size of the videos. I am wondering whether the authors also consider open-sourcing the collected data?**
>
> **A3**: Thanks for your suggestion. We construct our training corpus by combining both real and synthetic videos to cover diverse motion patterns. For the real videos, we directly collect footage from real-world sources. For the synthetic videos, we use Wan2.1-14B-I2V to generate videos containing a wide range of motion types but without control signals. In total, we gather approximately 10,000 videos. We then generate trajectory-based control signals using an automatic detector, followed by manual filtering to remove videos containing sensitive content, low-quality samples, or incorrect trajectories. For challenging scenarios, such as severe occlusion or fast motion, we curate a high-quality subset of approximately 3,000 videos, all of which are fully annotated by human annotators. We have clarified our data curation process in Appendix B.1.
>
> We will open-sourcing our dataset and the full processing pipeline after the paper is published to facilitate future research.
>
> ------
>
> **W4. Considering the core design of the self-rollout is very similar to self-forcing, I would suggest adding a section on a side-by-side comparison of the implementation levels (like pseudo-code) to highlight the implementation differences.**
>
> **A4**:  Thanks for your suggestion. We need a base model obey the Markov Decision Chain of AR video generation before RL training. However, Self-Forcing skips steps for every frame during training, which breaks the Markov Decision Chain. We propose Self-Rollout, which strictly enforces the Markov property. We have provided a detailed comparison between Self-Rollout and Self-Forcing, along with the algorithm and pseudo-code for both methods in Appendix A.
>
> ------
>
> Thank you again for your constructive reviews. Hope that our response can address your concerns. We will feel grateful if you could boost our paper.

---

> ### Author Response · Authors · 2025-11-28
> **Expecting to receive further feedback**
>
> Dear Reviewer RqLj,
>
> Thank you once again for your thoughtful and constructive feedback. We have carefully addressed each of your comments with detailed responses, conducted additional experiments, and provided more in-depth analysis based on your latest suggestions. We have also uploaded a revised version of the paper with all modifications clearly highlighted.
>
> Your insights have been invaluable in helping us strengthen our work, and we would greatly appreciate it if you could let us know whether there are any remaining issues or points that may require further clarification. We would be more than happy to provide any additional explanation you may need.
>
> Thank you again for your time and valuable input. We look forward to hearing from you.
>
> Best regards,
>
> The Authors

---

### Official Review · Reviewer_Zmb3 · 2025-10-30

**Soundness:** 2
**Presentation:** 2
**Contribution:** 2
**Rating:** 4
**Confidence:** 4

**Summary:**

AR-Drag is a novel real-time video generation system that addresses the latency limitations of traditional diffusion models by introducing the first reinforcement learning-enhanced autoregressive approach for image-to-video generation with motion control. The method fine-tunes a base model to support motion control and then improves it through reinforcement learning with a trajectory-based reward model, utilizing a Self-Rollout mechanism to preserve the Markov property and selective stochasticity to accelerate training. With only 1.3B parameters, AR-Drag achieves high visual quality and precise motion alignment while significantly reducing generation latency compared to existing state-of-the-art motion-controllable video diffusion models, overcoming the quality degradation and motion artifacts common in few-step generation approaches.

**Strengths:**

1. This paper introduces the first RL-enhanced autoregressive video diffusion model that enables real-time motion-controllable generation with significantly reduced latency compared to existing bidirectional approaches.
2. This paper proposes a Self-Rollout mechanism that preserves the Markov property while selectively introducing stochasticity in denoising steps, enabling efficient reinforcement learning training for few-step video generation.
3. This paper achieves high visual fidelity and precise motion alignment with only 1.3B parameters, overcoming the quality degradation and motion artifacts that typically plague few-step autoregressive video generation methods.

**Weaknesses:**

1. Compared to the pretrained model used, the video quality provided in the supplementary materials of this paper shows significant degradation with very obvious artifacts. For example, there are very obvious artifacts in the cat's pants in the first video.
2. The authors only provide very few videos in the supplementary materials, and none of these videos are annotated with motion control.
3. The authors do not discuss the limitations of this method in the paper. Please ask the authors to supplement this in the rebuttal.

**Questions:**

Please refer to weaknesses.

---

> ### Author Response · Authors · 2025-11-23
> **Feedback to Reviewer #Zmb3**
>
> We appreciate your overall positive assessment of our contributions and are grateful for your suggestion. We have already updated the new revision based on your suggestions.
>
> ------
>
> **W1. Compared to the pretrained model used, the video quality provided in the supplementary materials of this paper shows significant degradation with very obvious artifacts.**
>
> **A1**: We respectfully disagree with the reviewer’s assessment regarding artifacts. **These artifacts originate from the inherent limitations of the Wan2.1 pretrained backbone rather than our method**. Wan2.1 has only 1.3B parameters, and for our task—which involves fine-grained details and highly flexible motion-control signals—it is expected that our model may exhibit slight degradation.
>
> We compare our model against competitive baselines, the pretrained backbone, and the fine-tuned teacher model in our project website in Section 2. Our method is distilled from the Wan2.1. The fine-tuned teacher model is a 50-step bidirectional VDM, while our model is a 3-step AR VDM. The fine-tuned teacher model can be regarded as an approximate upper bound, and our results achieve **comparable quality** to it.  Moreover, thanks to the RL post-training, our model even achieves better motion consistency, surpassing the teacher model.
>
> In contrast, Self-Forcing, which is **also distilled from the teacher model, suffers from severe quality degradation**. This clearly highlights the effectiveness of our design.
>
> ------
>
> **W2. The authors only provide very few videos in the supplementary materials, and none of these videos are annotated with motion control.**
>
> **A2**: Thanks for your suggestions. We have added more visualization results in our project page. To make the controllability clearer, we now also include annotations of the motion control signals for every example.
>
> Our project page is organized into three sections.
> Section 1 compares AR-Drag with Tora, MagicMotion, and Self-Forcing. Tora and MagicMotion fail under fine-grained, highly flexible control, while Self-Forcing exhibits severe quality degradation.
> Section 2 compares AR-Drag with Self-Forcing, the pretrained backbone, and the fine-tuned teacher model. Self-Forcing again shows strong degradation, and the pretrained backbone struggles with motion consistency. AR-Drag achieves **quality comparable to the fine-tuned teacher model**—which serves as an approximate upper bound—and, thanks to RL post-training, even surpasses it in motion consistency.
> Section 3 demonstrates streaming generation with multiple control inputs. AR-Drag naturally supports streaming input, enabling better temporal consistency and higher visual quality. Bidirectional models (the pretrained backbone and the teacher model) cannot perform streaming generation, so for comparison we feed all control signals jointly when comparing against them.
>
> ------
>
> **W3. The authors do not discuss the limitations of this method in the paper. Please ask the authors to supplement this in the rebuttal.**
>
> **A3**: Thanks for your suggestions. Our generative model is trained on data that follows physical plausibility, and our reward model is also designed to evaluate motion based on physical principles. Therefore, if a user intentionally provides highly exaggerated or physically impossible control signals, the model may not strictly follow such inputs because they fall outside the distribution it is trained and rewarded to respect. Handling deliberately non-physical or cartoon-like motion is an interesting direction for future work, and we believe extending controllability beyond physically plausible dynamics is a valuable avenue for exploration.
>
> We have added the discussion of our limitation in Section 5.
>
> ------
>
> Thank you again for your constructive reviews. Hope that our response can address your concerns. We will feel grateful if you could boost our paper.

---

> ### Author Response · Authors · 2025-11-28
> **Expecting to receive further feedback**
>
> Dear Reviewer Zmb3,
>
> Thank you once again for your thoughtful and constructive feedback. We have carefully addressed each of your comments with detailed responses, conducted additional experiments, and provided more in-depth analysis based on your latest suggestions. We have also uploaded a revised version of the paper with all modifications clearly highlighted.
>
> Your insights have been invaluable in helping us strengthen our work, and we would greatly appreciate it if you could let us know whether there are any remaining issues or points that may require further clarification. We would be more than happy to provide any additional explanation you may need.
>
> Thank you again for your time and valuable input. We look forward to hearing from you.
>
> Best regards,
>
> The Authors

---

### Official Review · Reviewer_4Dc7 · 2025-10-31

**Soundness:** 3
**Presentation:** 3
**Contribution:** 3
**Rating:** 6
**Confidence:** 3

**Summary:**

The authors propose a real-time video generator using autoregressive diffusion model, named AR-Drag. To achieve it, they first finetune a I2V model to achieve motion control. Then the model adopts post-training via reinforcement learning with a trajectory-based reward model. It achieves the state-of-the-art performance using only 1.3B parameters.

**Strengths:**

1. Real-time video generation is a crucial topic in the video generation community. AR-Drag proposes a valid solution to it.
2. The latency analysis shows AR-Drag is close to real-time generation. The quantitative metrics demonstrates the method achieves the SOTA performance in trajectory controlled video generation.
3. The method incorporates a GRPO post-training process that further improves the performance.

**Weaknesses:**

1. An important baseline is missing. Please compare AR-Drag to MagicMotion.
2. Can you compare AR-Drag with Longlive[2], which is alose a real-time video generation with 1.3B parameters.

**Questions:**

Please see the weaknesses.

---

> ### Author Response · Authors · 2025-11-23
> **Feedback to Reviewer #4Dc7**
>
> We appreciate your overall positive assessment of our contributions and are grateful for your suggestion. We have already updated the new revision based on your suggestions.
>
> ------
>
> **W1. Please compare AR-Drag to MagicMotion.**
>
> **A1**: Thanks for your suggestion, we have added the baseline, Magicmotion, which is a bidirectional method with two versions: one fine-tuned from Wan2.1-1.3B-I2V and the other from CogVideoX-5B. However, only the 5B version is publicly released. The results are shown in Table 1 and Figure 3.
>
> AR-Drag consistently outperforms Magicmotion across all metrics, particularly on motion-control metrics. Magicmotion does not incorporate RL training and therefore cannot enable fine-grained or highly flexible motion control. Moreover, due to its bidirectional architecture, Magicmotion suffers from **extremely high latency (1426.37s)**, making real-time generation infeasible.
>
> ------
>
> **W2. Can you compare AR-Drag with Longlive, which is also a real-time video generation with 1.3B parameters.**
>
> **A2**: For Longlive, it has not been formally published yet. It was submitted to arXiv after our submission. Moreover, LongLive is a **T2V model** that focuses on coarse-grained text-prompt control, while our work focuses on fine-grained and highly flexible control for the I2V setting. Nevertheless, we believe the techniques in LongLive may be complementary to ours for future extensions involving text-based control.  We have added this related work in Section 2.
>
> ------
>
> Thank you again for your constructive reviews. Hope that our response can address your concerns. We will feel grateful if you could boost our paper.

---

> ### Author Response · Authors · 2025-11-28
> **Expecting to receive further feedback**
>
> Dear Reviewer 4Dc7,
>
> Thank you once again for your thoughtful and constructive feedback. We have carefully addressed each of your comments with detailed responses, conducted additional experiments, and provided more in-depth analysis based on your latest suggestions. We have also uploaded a revised version of the paper with all modifications clearly highlighted.
>
> Your insights have been invaluable in helping us strengthen our work, and we would greatly appreciate it if you could let us know whether there are any remaining issues or points that may require further clarification. We would be more than happy to provide any additional explanation you may need.
>
> Thank you again for your time and valuable input. We look forward to hearing from you.
>
> Best regards,
>
> The Authors

---

### Official Review · Reviewer_7uAx · 2025-11-02

**Soundness:** 3
**Presentation:** 2
**Contribution:** 2
**Rating:** 4
**Confidence:** 5

**Summary:**

This paper presents AR-Drag, a real-time, motion-controllable autoregressive (AR) video diffusion model optimized for I2V generation. It considers two challenges in AR video diffusion: the quality degradation and motion artifacts due to error accumulation and limited support for complex motion control. A two-stage training pipeline is proposed. The first stage is the base model training, including distilling the model into a few-step AR model, coupled with self-rollout, a training strategy that aligns training with inference. The second stage is reinforcement learning enhancement, which use GRPO to optimize the motion fidelity and visual quality via a trajectory-based reward.

**Strengths:**

1. The model has the real-time inference capability, orders of magnitude faster than bidirectional models, and achieved low FID and FVD score.

2. This work applies GRPO to AR video diffusion, introducing a trajectory-based reward for quality improvement.The

**Weaknesses:**

1. The novelty is quite limited, as the core components, like distilling, self-rollout, GRPO for video generation, have been individually explored in prior works. The proposed self-rollout only improves from self-forcing with minor modifications. The selective stochasticity adopted by AR-Drag that only chooses one denoising step instead of the whole stochastical trajectory is also a simple improvement.

2. The comparison in Table 1 is not that fair. AR-Drag uses the Wan backbone, which is much better than the backbone adopted in some competitor methods, line DrageNUWA, Tora. For Self-Forcing, it is not specifically designed for I2V (it’s backbone is T2V).

3. I didn’t find the detailed description of the conditioning on the previous frames (i.e. how the AR works). Only a figure is presented.

**Questions:**

Please see the weaknesses.

---

> ### Author Response · Authors · 2025-11-23
> **Feedback to Reviewer #7uAx**
>
> We appreciate your overall positive assessment of our contributions and are grateful for your suggestion. We have already updated the new revision based on your suggestions.
>
> ------
>
> **W1. Distilling, self-rollout, GRPO for video generation, have been individually explored in prior works. The proposed self-rollout and selective stochasticity sampling improve with minor modifications.**
>
> **A1**: We respectfully disagree with the reviewer’s assessment regarding novelty. While prior works have individually explored distillation, rollout strategies, or GRPO, none of them address the fundamental obstacles that make real-time, controllable, autoregressive video generation feasible. Specifically, existing methods do not (1) restore the Markov property required for RL in AR VDMs, (2) control the exploding variance inherent in extremely long AR denoising chains, or (3) provide reward models suitable for fine-grained motion control. Our core contribution is a principled framework that resolves these challenges and, to the best of our knowledge, **enables the first real-time, controllable AR video diffusion model.** Specifically,
>
> **(1) No prior work applies GRPO to motion-controllable AR video generation, because the setting is fundamentally challenging due to its non-Markov nature.** Typical AR VDMs suffer from a severe training-inference gap: during inference, the model generates the current frame conditioned on previously generated frames, whereas during training it is conditioned on ground-truth frames. This breaks the Markov property and makes policy-gradient methods such as GRPO theoretically invalid.  To make RL feasible, we first fine-tune Wan2.1-I2V into a base model that strictly satisfies the Markov decision process.  While Self-Forcing partially reduces the mismatch by updating the KV cache with generated frames, it **skips steps for every frame, which breaks the Markov Decision Chain**. In contrast, we propose Self-Rollout, which strictly enforces the Markov property. conceptually simple, this modification is essential—it **completes the MDP formulation**,  enables the use of GRPO, and leads to **substantial performance improvements** over Self-Forcing (see comparisons with Self-Forcing in Table 1 and Figure 3, and ablation study on Self-Rollout in Table 2 and Figure 4). We also provide a detailed comparison between Self-Rollout and Self-Forcing, along with the algorithm and pseudo-code for both methods, in Appendix A.
>
> **(2) Extremely long decision chains make motion-controllable AR video generation fundamentally difficult for GRPO.** In bidirectional VDMs, all frames are denoised jointly, so the decision horizon is only the number of denoising steps. In contrast, AR VDMs generate frames sequentially, making the horizon proportional to the **product of the number of frames and denoising steps** (often hundreds to thousands of decision steps). Such extremely long horizons cause severe return-variance explosion, destabilizing policy-gradient estimates and drastically reducing sample efficiency. This is precisely why directly applying GRPO (or any standard policy-gradient method) to AR video generation is infeasible in practice. To address this core difficulty, we introduce **selective stochasticity sampling**, where only a single randomly chosen denoising step uses stochastic SDE updates and all other steps remain deterministic. This strategy injects sufficient exploration for GRPO while **avoiding variance blow-up**, enabling stable optimization with practical rollout budgets. As shown in our experiments, this design is critical for making GRPO effective in the AR video generation setting.
>
> **(3) No existing reward model is available for fine-grained controllable video generation.** We employ Co-Tracker to measure the alignment of the generated frames with ground-truth frames successfully.
>
> In summary, our modifications are **simple yet effective**, directly addressing the challenges in applying GRPO to AR VDMs. We have clarified our technical novelty more clearly in Section 5.

---

> ### Author Response · Authors · 2025-11-23
> **Feedback to Reviewer #7uAx**
>
> **W2. The comparison in Table 1 is not that fair. AR-Drag uses the Wan backbone, which is much better than the backbone adopted in some competitor methods. For Self-Forcing, it is not specifically designed for I2V.**
>
> **A2**: To the best knowledge of us, there is no open-sourced Wan model  capable of **real-time, fine-grained, and highly flexible controllable image-to-video (I2V) generation**. We are the first one.
>
> Previous controllable video generation models rely on older backbones, and existing Wan-based models are not designed for controllable video generation. They can only support **simple interactions**, such as pose-based or camera-style control derived from the Wan model, which is **insufficient for fine-grained and highly flexible controllability** in I2V generation. Furthermore, these models are not autoregressive and therefore cannot support real-time generation.
>
> To enable motion controllability in an AR setting, we fine-tune an Wan-based AR VDM, Self-Forcing. Self-Forcing only releases their t2v version. So we fine-tune Wan2.1-1.3B-I2V following the Self-Forcing architecture and training procedure using the same datasets as AR-Drag.
>
> Thanks for Reviewer #4Dc7's suggestion, we also add a new baseline, Magicmotion, which is a bidirectional method with two versions: one fine-tuned from Wan2.1-1.3B-I2V and the other from CogVideoX-5B. However, only the 5B version is publicly released. The results are shown in Table 1 and Figure 3. AR-Drag consistently outperforms Magicmotion across all metrics, particularly on motion-control metrics. Magicmotion does not incorporate RL training and therefore cannot enable fine-grained or highly flexible motion control. Moreover, due to its bidirectional architecture, Magicmotion suffers from **extremely high latency (1426.37s)**, making real-time generation infeasible.
>
> We have updated the detailed descriptions of our baselines in Section 4, and added the new baseline in Table 1 and Figure 4 (Section 4.1).
>
> ------
>
> **W3. I didn’t find the detailed description of the conditioning on the previous frames (i.e. how the AR works).**
>
> **A3**: Thanks for your suggestion. In Self-Rollout, the autoregressive conditioning on previous frames is implemented exactly as in inference, ensuring training–inference consistency. Specifically, each new frame is generated by denoising from pure Gaussian noise, conditioned on the KV cache accumulated from previously generated clean frames. Once a frame is fully denoised to the clean state, it is used to update the KV cache, which is then employed to condition the generation of subsequent frames.
>
> We have included the full Self-Rollout training process along with detailed pseudo-code in Alg. 1 and Listing 1 in Appendix A to more clearly demonstrate how the autoregressive process operates.
>
> ------
>
> Thank you again for your constructive reviews. Hope that our response can address your concerns. We will feel grateful if you could boost our paper.

---

> ### Author Response · Authors · 2025-11-28
> **Expecting to receive further feedback**
>
> Dear Reviewer 7uAx,
>
> Thank you once again for your thoughtful and constructive feedback. We have carefully addressed each of your comments with detailed responses, conducted additional experiments, and provided more in-depth analysis based on your latest suggestions. We have also uploaded a revised version of the paper with all modifications clearly highlighted.
>
> Your insights have been invaluable in helping us strengthen our work, and we would greatly appreciate it if you could let us know whether there are any remaining issues or points that may require further clarification. We would be more than happy to provide any additional explanation you may need.
>
> Thank you again for your time and valuable input. We look forward to hearing from you.
>
> Best regards,
>
> The Authors

---

### Author Response · Authors · 2025-11-23
**General response**

We thank all reviewers for their insightful comments and suggestions.  We are encourage that they found our paper clearly written and well-designed (#RqLj), technically sound with an effective Self-Rollout framework and smart selective stochasticity sampling (#Zmb3, #RqLj), and supported by strong empirical results (#7uAx, #4Dc7, #Zmb3). We appreciate their recognition that our method enables real-time generation with state-of-the-art visual and motion quality despite using only a 1.3B-parameter model (#7uAx, #4Dc7, #Zmb3).

We have made a revision to our paper according to the reviewer's constructive suggestions. Below we summarize the key modifications in this revision:

- **Clarified technical contributions.** We added a dedicated discussion of the two main components, Self-Rollout and selective stochasticity sampling, to better highlight our contribution and technical novelty (Section 5; #7uAx).
- **Expanded baseline settings.** We added detailed descriptions of all baseline configurations (Section 4; #7uAx) and included a new baseline, MagicMotion (Section 4.1; #4Dc7).
- **Added algorithms and pseudo-code.** We provided full algorithmic details and pseudo-code illustrating how our AR framework operates (#7uAx), as well as a clear comparison between Self-Rollout and Self-Forcing (Appendix A; #RqLj).
- **Added more qualitative results.** We expanded the project page with additional examples and comparisons against baselines, the pretrained backbone, and the fine-tuned teacher model (#Zmb3, #RqLj).
- **Added limitation discussion.** We included a discussion of our limitations in Section 5 (#Zmb3).

If there still remains any consideration, Please kindly let us know. We are very happy to make a further revision in light of your great suggestions. We will address comments by each of the reviewers individually.

---

### Author Response · Authors · 2025-12-03
**Rebuttal Summary for Newly Assigned AC**

Dear re-assigned AC,

We sincerely appreciate your tremendous efforts in handling the urgent OpenReview information-leak incident! **To speed up your review process, we would like to provide some critical information about our submission**.

------

**All the four reviewers `consistently acknowledge the performance and contribution` of our work.** For example,

1. `Reviewer 7uAx`: "has the real-time inference capability", "achieved low FID and FVD score", "applies GRPO to AR video diffusion".
2. `Reviewer 4Dc7`: " a valid solution", "incorporates a GRPO post-training process".
3. `Reviewer Zmb3`: " achieves high visual fidelity and precise motion alignment", "enabling efficient reinforcement learning training for few-step video generation".
4. `Reviewer RqLj`: " has a clean, straightforward design", "The modification of self-forcing is a smart design".

------

In the first-round review, `Reviewer 4Dc7 and RqLj`  provided positive scores, recognizing both the strong performance and the contributions of our work.  While `Reviewer 7uAx and Zmb3`, however, raised several concerns and suggestions. Due to the information-leak incident, the reviewers were unable to comment on our rebuttal in time. Below, we summarize the main concerns from `Reviewer 7uAx and Zmb3` as well as our corresponding responses.

`Reviewer 7uAx`

- **Technical details and novelty**: We have added detailed algorithm and pseudo code in Appendix A to illustrate the differnce of our method and related approach, which also clarifies our novelty.
- **Baseline setting**: We clarified our baseline setup and included an additional baseline in Section 4.1.

`Reviewer Zmb3`

- **More visualization results and campared with pretrained model**: We expanded the project page with additional examples and comparisons against baselines, the pretrained backbone, and the fine-tuned teacher model.
- **Limitation discussion of our method**: We added a discussion of our method’s limitations in Section 5.

------

Thank you again for your time and efforts.

Best regards,

Authors #8617

---

### Meta-Review · Area_Chair_fQHx · 2026-01-07

**Summary:**

The paper presents a promising advancement in the field of motion-controllable video generation, particularly by enabling real-time inference using autoregressive diffusion models. Despite some concerns regarding novelty and baseline fairness, the authors have addressed these points adequately through revisions and clarifications.

Given the substantial improvements in motion control, generation quality, and latency reduction, as well as the authors’ clear and comprehensive responses to reviewer concerns, I recommend the paper for acceptance, provided that the remaining minor issues regarding the baseline fairness, data transparency, and supplementary material quality are adequately addressed.

**Reviewer Concerns:**

- Reviewer 7uAx concerned with the novelty of the approach due to prior work on distillation, self-rollout, and GRPO. Also questioned the fairness of comparisons with other models, especially regarding backbone choices and lack of detailed conditioning description. The authors clarified their contributions and addressed the concerns by adding new baselines and providing more detailed explanations.

- Reviewer 4Dc7 suggested adding MagicMotion as a baseline and requested a comparison with LongLive. The authors responded by adding MagicMotion, showing AR-Drag’s superior performance, and explaining the differences with LongLive, noting it focuses on text-based control.

- Reviewer Zmb3 raised concerns about video quality degradation and artifacts in the supplementary materials and the lack of motion control annotations. Also requested a discussion on the model’s limitations. The authors clarified the artifacts were due to the pretrained backbone, updated the supplementary materials, and discussed the model’s limitations in handling non-physical inputs.

- Reviewer RqLj pointed out the incomplete website, lack of baseline comparisons, and motion trajectory visualizations. Also questioned the fairness of optimizing FID/FVD metrics directly during training. The authors addressed these by improving the project page and clarifying that their improvements are validated through both quantitative and qualitative analysis. Additionally, they provided more details about the training data and their curation process.

**Reviewer Scores:**

The scores from the reviewers are somewhat split, with two reviewers providing ratings of 4, and two with 6. Concerns were raised about the paper's novelty, fairness in baseline comparisons, clarity of the presented details, and some visual artifacts. However, the authors made significant revisions, addressing most of these points by clarifying the novelty, adding additional baselines, improving supplementary materials, and discussing the limitations of their approach. Although some concerns about scalability and generalization persist, the paper has addressed the majority of reviewer issues.

---

### Decision · Program_Chairs · 2026-01-26

Accept (Poster)